# Psychosocial supports within pediatric nephrology practices: A pediatric nephrology research consortium survey

**Anne E. Dawson**[1,2]*, **Camille S. Wilson**[1,2], **William E. Smoyer**[2,3], **Neha Pottanat**[4,5], **Amy C. Wilson**[4,5], **John D. Mahan**[2], **Julia E. LaMotte**[5,6]

**1** Department of Psychology and Pediatric Neuropsychology, Nationwide Children's Hospital, Columbus, OH, United States of America, **2** Department of Pediatrics, The Ohio State University, Columbus, OH, United States of America, **3** Center for Clinical and Translational Research, Nationwide Children's Hospital, Columbus, OH, United States of America, **4** Division of Nephrology, Riley Hospital for Children, Indianapolis, IN, United States of America, **5** Division of Hematology/Oncology, Riley Hospital for Children, Indianapolis, IN, United States of America, **6** Department of Pediatrics, Indiana University School of Medicine, Indianapolis, IN, United States of America

* anne.dawson@nationwidechildrens.org

**Data Availability Statement:** All relevant data are within the paper and its Supporting Information file.

## Abstract

### Background

The landscape of available psychosocial services within pediatric nephrology care is poorly characterized. However, the effects of kidney disease on emotional health and health-related quality of life are well documented, as is the impact of social determinants of health on kidney disease outcomes. The objectives of this study were to assess pediatric nephrologists' perceptions of available psychosocial services and to elucidate inequities in access to psychosocial care.

### Methods

A web-based survey was distributed to members of the Pediatric Nephrology Research Consortium (PNRC). Quantitative analyses were performed.

### Results

We received responses from 49 of the 90 PNRC centers. With regards to dedicated services, social work was most commonly available (45.5–100%), followed by pediatric psychology (0–57.1%) and neuropsychology (0–14.3%), with no centers having embedded psychiatry. Availability of psychosocial providers was positively associated with nephrology division size, such that as center size increased, access to various psychosocial providers increased. Notably, the majority of respondents indicated that perceived need for psychosocial support exceeds that which is currently available, even at centers with higher levels of current support.

**Funding:** The author(s) received no specific funding for this work.

**Competing interests:** The authors have declared that no competing interests exist.

**Abbreviations:** CKD, Chronic kidney disease; CKiD, Chronic Kidney Disease in Children; PNRC, Pediatric Nephrology Research Consortium; CMS, Centers for Medicaid and Medicare Services; RDU, Renal Dialysis Unit; HRQOL, Health-related quality-of-life.

## Conclusions

Within the US, there is wide variability in the availability of psychosocial services within pediatric nephrology centers despite a well-documented necessity for the provision of holistic care. Much work remains to better understand the variation in funding for psychosocial services and in utilization of psychosocial professionals in the pediatric nephrology clinic, and to inform key best practices for addressing the psychosocial needs of patients with kidney disease.

## Introduction

Pediatric nephrologists are responsible for providing complex and multifaceted medical care across diverse kidney disease groups in children. This is inclusive of acute kidney injury, chronic kidney disease (CKD), chronic kidney failure, congenital or genetic disorders with kidney involvement (e.g., polycystic kidney disease, congenital nephrotic syndrome, and hypo/dysplastic kidneys), glomerular disease, hypertension, kidney stones and urinary or urologic abnormalities. In addition to the breadth of diagnoses, the intricacies of disease management are multifaceted, including dietary and/or fluid modifications along with complex and strict medication regimens that when not followed closely may result in increased morbidity and mortality. Multisite research efforts, including the Chronic Kidney Disease in Children study (CKiD) have sought to better understand the course of mild to moderate impaired kidney function in CKD, including impacts on psychosocial and cognitive functioning [1–3]. Additionally, groups such as the Pediatric Nephrology Research Consortium (PNRC) have been established to improve and promote high quality care across pediatric nephrology centers, centered on the shared goal of increasing understanding and treatment of pediatric kidney disease. Given the heterogeneity of kidney disease and the chronicity of these conditions, some with a lifelong course, the medical and psychosocial burden of care remains a challenge due to the impact of these disorders on psychosocial development and growth in children, as well as highly variable disease presentation, course, and severity.

### Psychosocial concerns for kidney patients and families

Effects on psychological well-being have been reported across disease groups in pediatric nephrology. In addition to CKD, there are a range of psychosocial and cognitive consequences observed across other specific conditions with kidney involvement, including hypertension [4], prune belly syndrome [5], tuberous sclerosis complex [6], and nephrotic syndrome [7]. The psychosocial and cognitive outcomes associated with CKD are most widely studied among the diverse kidney conditions in pediatric patients. Indeed, a disproportionately high number of children (ages 9–18) with CKD meet diagnostic criteria for depression [8]. Patients with end stage kidney disease undergoing dialysis treatment report even greater rates of depression and anxiety, relative to both their peers without kidney disease and their peers who have pre-dialysis CKD [9, 10]. Over time, pediatric dialysis patients report worse emotional health [11] that often persists into adulthood [12]. Although medical advancements have led to improved long-term survival [13], the consequences of kidney disease sequelae and treatment in children clearly influence health-related quality-of-life (HRQOL). In a systematic review of pediatric dialysis patients, the impact on HRQOL was best understood across five themes [14]. These included *loss of control* (reliance on others, machine dependence) *restricted lifestyle*

(limited socialization), *coping strategies* (hopefulness towards transplant), *managing treatment* (adherence to dietary and fluid restrictions), and *feeling different* (the burden of perceived differences in physical appearance). Accordingly, both disease progression and medical treatments (e.g., dialysis) may affect children's self-esteem, personal identity, independence, and perceived control or agency. Due to elevated concerns for mental health problems and decreased quality of life in children with end stage CKD, the Centers for Medicaid and Medicare Services (CMS) requirements now mandate regularly screening these patients for depression and quality of life.

In addition to the effects on mental health, many patients with kidney diseases are also at increased risk of cognitive and academic dysfunction, which may be directly related to the disease process. For instance, the pathophysiologic effects of advanced uremia include impacts on brain metabolism and thereby cognitive functioning [15]. However, kidney replacement treatments may also affect cognitive functions [16]. Attending hemodialysis has a notable impact on limiting a child's ability to be physically present in the school environment, which not only has implications for academic achievement [2, 17], but also reduces opportunities for expected peer socialization and skill development. While some neurocognitive deficits may improve following kidney transplant, overall findings of children with CKD suggest lower intellectual functioning compared to those without CKD [3, 18, 19]. Patterns of difficulties include challenges with executive functions including holding information and shifting attention, reduced visual and verbal memory, and lower metacognitive skills [17].

With increasing awareness of the direct and indirect impacts of the medical and emotional burden that kidney disease can have on patient and family functioning, psychosocial professionals have emerged as important providers in kidney care. Increasingly, the importance of specialized training in nephrology for psychosocial professionals have been identified as imperative for maximizing support for patients and families. For example, the National Kidney Foundation Council of Nephrology Social Workers developed a Nephrology Social Worker Certification in 2009 [20], and the Society for Pediatric Psychology, a division of the American Psychological Association, began a special interest group with a focus in nephrology in 2018 [21]. Specialty training opportunities as well as professional interest groups reflect the need for systematic and coordinated care of patients with high levels of disease complexity and psychosocial needs. The COVID-19 pandemic may amplify the psychosocial issues inherent in kidney disorders, as children with chronic medical conditions such as pediatric nephrology patients are most at risk for experiencing a mental health crisis today [22, 23]. The US Surgeon General recently documented children have increasing rates of depression, suicidal ideation and attempts, and more challenges in relation to social determinants of health (mental health, social services, food, housing, and caregiver health); the vulnerabilities within the pediatric nephrology population have similarly also increased [23].

## Impact on the pediatric nephrology workforce

The perceived high complexity of patient needs, along with insufficient support for psychosocial services, may serve to dissuade medical residents from pursuing a career in pediatric nephrology [24]. Unfortunately, concerns about the pediatric nephrology workforce [24, 25] have only escalated [26]. Given the high complexity of care needs, it is possible that expansion of the psychosocial services available to meet holistic needs of patients and their families will reduce clinical burdens on nephrologists and lead to improvements in workforce growth. Indeed, given the complex needs of patients in the pediatric nephrology subspecialty, comprehensive disease management is best achieved through multidisciplinary care. Salerno, Weinstein, and Hanevold [27] supported this notion, outlining distinct personnel needs of pediatric

nephrology practices in addition to the nephrologist, including nurse specialists, dieticians, social workers, clinical administrators, psychologists, child life specialists, dialysis nurses, and renal transplant coordinators (for those with transplant programs). Through overlapping interests and unique contributions of each discipline, patients may experience a high level of integrated care that may also allay stressors experienced by individual providers. Indeed, adequate psychosocial support is a key recommendation in treatment for infants with kidney failure [28] as well as in treatment of children with hypertension [4]. Multidisciplinary care allows for management of comorbidities, focused on the patient's unique needs, and has implications for improved clinical outcomes [29]. Review of the nephrologist's perception of psychosocial care in hemodialysis units revealed that the overwhelming majority (94%) shared the belief that patient outcomes improve with focus on psychosocial care [30]. Additionally, while most (78%) nephrologists identified as participating in psychosocial initiatives, less than half (40%) lead these initiatives and very few (9%) received training in aspects of psychosocial issues, suggesting that while nephrologists report perceived benefit from interdisciplinary collaboration, the nephrologist and each psychosocial provider make unique contributions to patient care.

However, the solution is not as simple as having increased access to psychosocial professionals, but also involves careful consideration of division infrastructure and provider support. For example, in many kidney care settings with additional support, social workers are the primary designated psychosocial provider. Hansen and colleagues [31] documented the broad psychosocial support that social workers provide, ranging from psychoeducation, counseling for parents and families, and significant care coordination, such as linkage to community and educational resources. While these supports have been highlighted in the literature to be key factors to help address high rates of depression, lower quality of life, and family stressors associated with higher acuity kidney needs, the burden of who is responsible for addressing these psychosocial needs carries a cost [32, 33]. There is a notable body of research documenting high levels of burnout among social workers and nurses in dialysis centers [33]. Burnout may be hastened by the complex patient needs, high patient volume, and the vast responsibilities largely shouldered by social workers in pediatric nephrology centers. Identified barriers to providing highly valued psychosocial care to pediatric nephrology patients include lack of access to psychosocial healthcare providers, high administrative demands, lack of training in psychosocial care delivery, and empathy fatigue [30]. Therefore, methodical and deliberate integration and inclusion of psychosocial providers, given value of integrated care to patients as well as reduction of burden on other providers that may help decrease burnout and address ongoing workforce shortages.

While a multidisciplinary model has been detailed as the standard of care within pediatric nephrology [27], little is known about the implementation of these guidelines into pediatric nephrology practices in the United States. In the United Kingdom, review of their renal psychosocial workforce showed great variability in service provision, as well as differences in staffing models including disparate availability of psychosocial team members (e.g., social worker, psychologist, counselor) between centers [34]. To address this gap, we surveyed pediatric nephrology centers belonging to the PNRC in the United States, with the goal of gaining a better understanding of the present availability of psychosocial services, as well as elucidate present disparities in access to psychosocial care.

## Methods

### Procedures

This study was deemed to be exempt by sponsoring institutions' Human Research Protection Programs. Participants were supplied with information about the study, so their choice to

continue was implied consent. The PNRC is a voluntary consortium of academic pediatric nephrology practices (N = 90) representing more than 60% of pediatric nephrology centers in the US, including the majority of large US centers. Each registered member of the PNRC was contacted via email requesting completion of an electronic survey through a REDCap link [35, 36]. The email recommended that participants complete the survey at a division or section meeting to ensure only one survey was submitted per PNRC affiliated site. Participants received up to three reminders to complete the survey. Of the responses, five PNRC centers completed the survey twice; the first response submitted was deleted from dataset. Authors also collected data from respective program websites for all PNRC centers that did not respond to determine representative nature of responses collected. Data were analyzed using IBM SPSS Version 26. Due to the exploratory nature of the study, statistical analyses were largely descriptive, with appropriate test statistics used to determine significant differences (i.e., Chi-Square/ T-Tests).

## Participants

Respondents represented 49 unique PNRC affiliated pediatric nephrology practices across the United States of America, encompassing 56% of PNRC membership in North America (see Fig 1 for geographic distribution). When comparing responding versus non-responding PNRC centers, there were no significant differences in size of centers, number of physicians (M.D. or D.O. providers), or midlevel professionals (N.P. or P.A. providers). Non-responding centers were less likely to have a renal dialysis unit (RDU; 70.2% vs. 92.1%; $X^2(1) = 6.294$, $p = .012$). There were no significant differences in percent of centers that conducted renal transplants between responding and non-responding PNRC centers. Of the top 40 pediatric nephrology centers ranked by the U.S. News and World Report's Best Children's Hospitals in

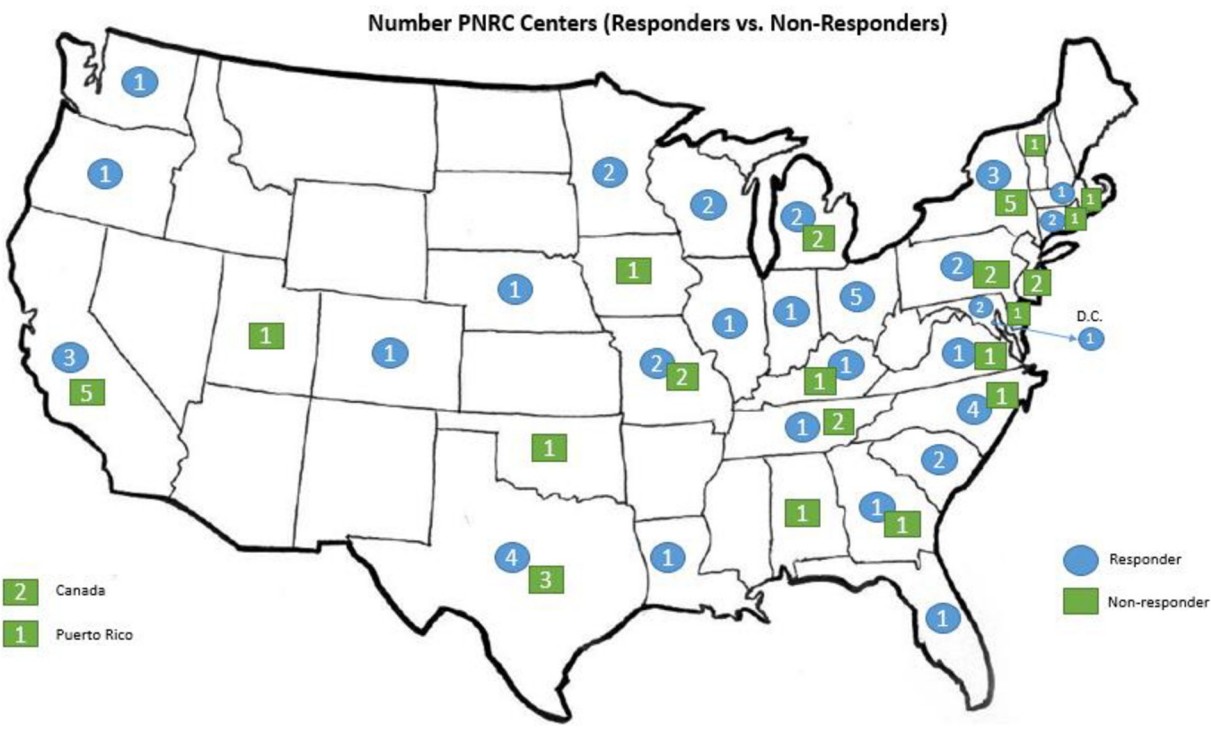

**Fig 1. National distribution of PNRC-affiliated pediatric nephrology centers.**

2021 [37], 34 centers participate in the PNRC; 26 of those centers provided survey responses in the current study.

## Measures

The survey was developed by the authors and revised based on recommendations made by the PNRC study approval committee. The survey included three parts; the first part collected institution characteristics (i.e., PNRC institution, number of providers, dialysis offered, number of chronic dialysis patients treated, type of kidney conditions treated, and whether transplants are performed). The second part assessed availability of psychosocial services (e.g., assigned to division, available for consultation, assigned to division, or not available), use of behavioral screeners, and the allied health professional/psychosocial services that are available to patients receiving chronic dialysis. The third part related to satisfaction, importance, and effectiveness of psychosocial services to meet the center's patient population needs.

## Results

### Institution characteristics

The number of nephrology providers per responding center was highly variable (physicians ranged from 1 to 21, $M = 6.12$; midlevel professionals ranged from 0 to 10, $M = 1.60$). The number of attending nephrologists per institution was used to categorize centers by size for purposes of statistical analysis. Corresponding to 25th, 50th, and 75th percentile values, small centers are classified as those having $\leq 3$ nephrologists, medium centers 4–7 nephrologists, and large centers $\geq 8$ nephrologists. For additional characteristics of centers by size, see Table 1. All centers indicated treating a wide variety of kidney conditions; however, there was pronounced variability in treatment of metabolic bone disease, with five centers (2 small, 1 medium, and 2 large) not treating these conditions within the nephrology division. Most centers performed kidney transplants, with the likelihood of this increasing with the size of the center $X^2(2) = 7.964$, $p = .019$. The total number of providers at a center (physicians plus midlevel professionals) was strongly associated with the number of chronic dialysis patients cared for at the center, $r = .738$, $p < .001$.

### Psychosocial services

Availability of psychosocial providers varied based on nephrology division size, such that as nephrology center size increased, access to various psychosocial providers increased. See Table 2 for availability of psychosocial services to entire nephrology divisions, as well as the variety of psychosocial services to renal dialysis units specifically. Further analysis indicated

**Table 1. Basic characteristics of pediatric nephrology programs by center size.**

|  | Small Centers (N = 11) | Medium Centers (N = 24) | Large Centers (N = 14) |
|---|---|---|---|
|  | *Mean (Range)* | *Mean (Range)* | *Mean (Range)* |
| MD/DO | 2.36 (1–3) | 5.08 (3.35–7) | 10.86 (8–21) |
| NP/PA | 0.27 (0–1) | 1.23 (0–3) | 3.21 (1–10) |
| # of chronic dialysis patients | 6.90 (0–17) | 13.88 (2–40) | 25.71 (9–49) |
|  | % (*N*) | % (*N*) | % (*N*) |
| RDU at institution | 45.5% (5) | 70.8% (17) | 78.6% (11) |
| Kidney Transplants performed | 63.6% (7) | 95.8% (23) | 100% (14) |

**Table 2. Access to psychosocial services by pediatric nephrology division and renal dialysis unit.**

| | Small Centers (N = 11) | Medium Centers (N = 24) | Large Centers (N = 14) |
|---|---|---|---|
| **Social Work** | | | |
| Not available | 0% | 0% | 0% |
| Can access/consult | 54.5% (N = 6) | 16.7% (N = 4) | 0% |
| Assigned to Division | 45.5% (N = 5) | 83.3% (N = 20) | 100% (N = 14) |
| **Pediatric Psychology** | | | |
| Not available | 0% | 0% | 0% |
| Can access/consult | 100% (N = 11) | 83.3% (N = 20) | 42.9% (N = 6) |
| Assigned to Division | 0% | 16.7% (N = 24) | 57.1% (N = 8) |
| **Neuropsychology** | | | |
| Not available | 18.2% (N = 2) | 8.3% (N = 2) | 0% |
| Can access/consult | 72.7% (N = 8) | 87.5% (N = 21) | 78.6% (N = 11) |
| Assigned to Division | 0% | 0% | 14.3% (N = 2) |
| **Psychiatry** | | | |
| Not available | 0% | 4.2% (N = 1) | 0% |
| Can access/consult | 100% (N = 11) | 95.8% (N = 23) | 100% (N = 14) |
| Assigned to Division | 0% | 0% | 0% |
| **Additional Dialysis Unit Specific Psychosocial Resources** | **Small Centers** (N = 5) | **Medium Centers** (N = 17) | **Large Centers** (N = 11) |
| School Teacher | 40.0% (N = 2) | 52.9% (N = 9) | 81.8% (N = 9) |
| School Liaison | 40.0% (N = 2) | 11.8% (N = 2) | 45.5% (N = 5) |
| Early Intervention | 40.0% (N = 2) | 5.9% (N = 1) | 27.3% (N = 3) |
| Child Life | 100.0% (N = 5) | 76.5% (N = 13) | 100.0% (N = 11) |
| Therapeutic Recreation | 0% (N = 0) | 5.9% (N = 1) | 36.4% (N = 4) |
| Music Therapist | 40.0% (N = 2) | 47.1% (N = 8) | 81.8% (N = 9) |
| Art Therapist | 20.0% (N = 1) | 23.5% (N = 4) | 54.5% (N = 6) |
| Massage Therapist | 0% (N = 0) | 0% (N = 0) | 18.2% (N = 2) |
| Nurse Educator | 40.0% (N = 2) | 35.3% (N = 6) | 63.6% (N = 7) |

**Note.** If percent does not add up to 100%, the remainder of respondents selected "I don't know." Additionally, respondents were asked if other services were available to their patients on consult basis or assigned to their division, and a total of 3 medium size centers and 4 large centers identified other resources, including: music therapy, quality of life specialists, palliative care providers, integrative pain providers, massage therapy, acupuncture, and pet therapy. The center size (N) decreases with "Dialysis Unit Specific" considerations as it only includes centers that indicated having an RDU at their institution.

that the larger the center, the more likely it was to have a dedicated social worker assigned to the division [$X^2(2) = 11.688$, $p = .003$] and the same was true for pediatric psychology [$X^2(2) = 12.434$, $p = .002$]. Likewise, responding centers that were ranked in top 40 of U.S. News and World Reports were more likely to have a pediatric psychologist specifically assigned to their nephrology division than those that were not [$X^2(1) = 5.841$, $p = .016$], but were not more likely to have an assigned social worker [$X^2(1) = 2.683$, $p = .101$]. Moreover, centers were more likely to have a social worker [$X^2(1) = 5.544$, $p = .019$] assigned to their division if they had a renal dialysis unit at their institution, but the same was not true for pediatric psychologists assigned to division [$X^2(2) = 2.941$, $p = .086$]. Available services in the RDU are also reported in Table 2 and did not significantly differ by center size. However, among centers with an RDU, those that were ranked in top 40 of U.S. News and World Reports were more likely to have school teachers [$X^2(1) = 6.310$, $p = .012$] and therapeutic recreation specialists [$X^2(1) = 4.342$, $p = .037$] as part of the psychosocial care team in their RDU than those that were not.

The majority of respondents (60.4%) reported that health screens are performed as a part of standard of care. Further analyses indicated that the majority centers administer behavioral

health screens to chronic dialysis patients (57.1% of centers administer to hemodialysis patients; 53.1% to peritoneal dialysis patients) and renal transplant patients (57.1%). Interestingly, centers ranked in the top 40 of pediatric nephrology centers by the U.S. News and World Report were more likely to administer behavioral health screens then other centers, $X^2(2) = 8.437$, $p = .015$. Relative to psychology supports specifically, respondents were asked to estimate how many of their patients were followed by psychology in some capacity. Although several respondents did not know (20.4%), the majority (63.3%; $n = 31$) reported that only 0–25% of their patients were followed by psychology.

## Perception of psychosocial services

Respondents were further queried as to the perceived importance of psychosocial supports in helping provide care in their nephrology division. Responses were largely favorable, with all indicating that psychosocial services were at least "somewhat important." As demonstrated in Fig 2, perceived importance increases with center size, $r = .405$, $p = .005$.

In addition to being asked to estimate how many of their current patients are followed by psychology, respondents were also asked how many of their patients would be followed by psychology in an "*ideal world.*" Although a small number (4.1%) indicated they did not know, the majority of respondents (55.1%) reported that ideally more than half of their patients would be followed by psychology. Although no respondents reported that ≥75% of their patients are currently followed by psychology, nearly a quarter of respondents (24.5%, $n = 12$) reported that in an ideal world 75% would indeed be seen by a psychologist. Fig 3

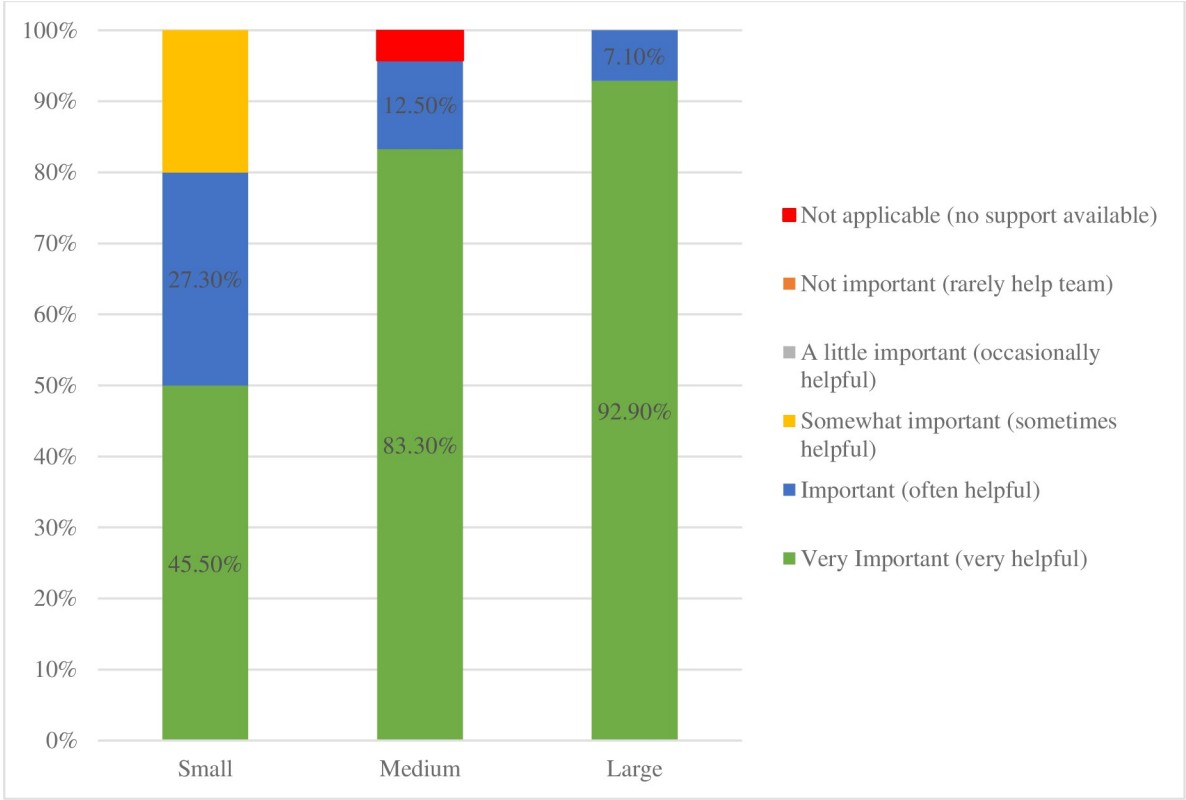

**Fig 2. Importance of psychosocial supports to the nephrology.**

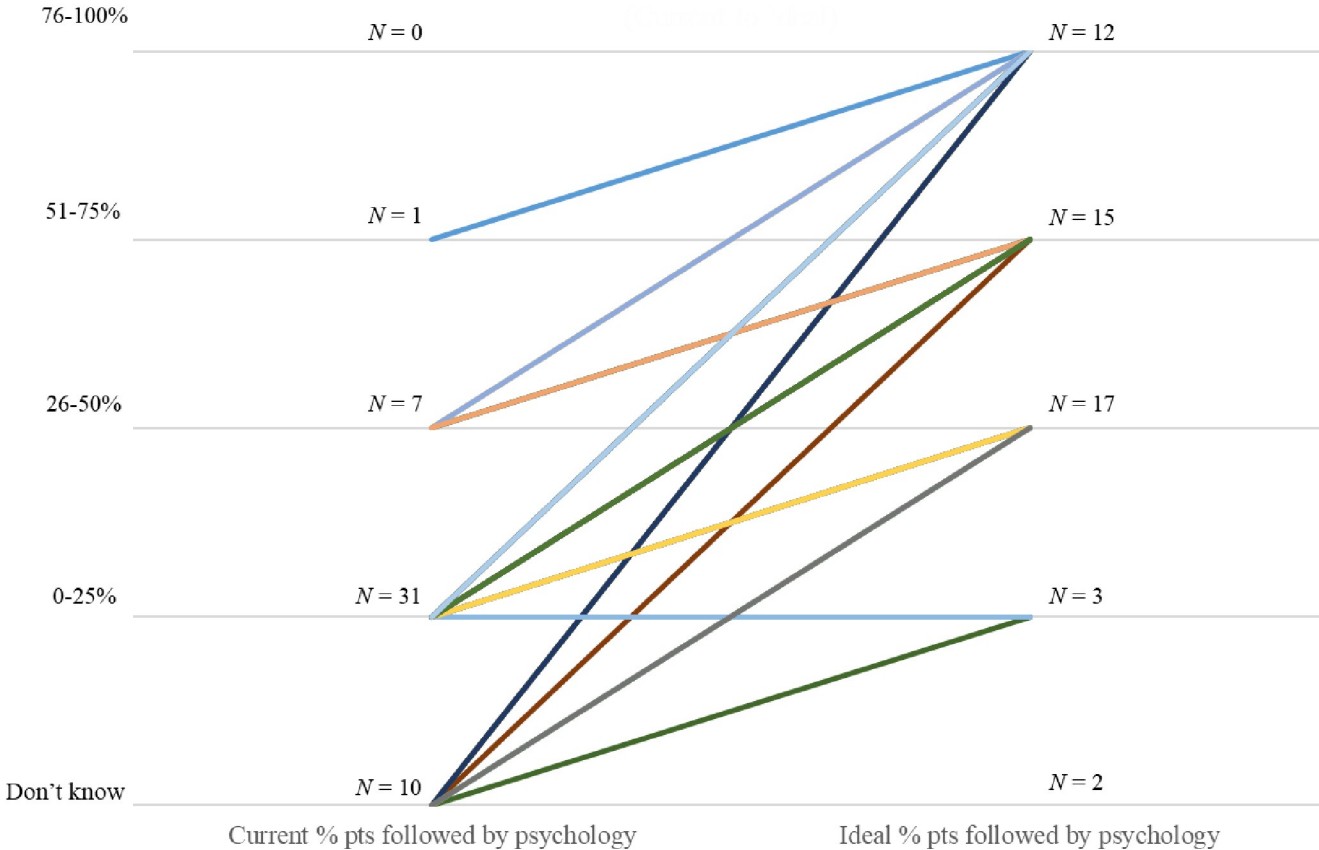

**Fig 3. Nephrology patients followed by psychology, current state versus ideal state.**

displays the reported current percentage of patients followed by psychology versus the percent of patients that would be followed by psychology in an ideal world for each responding center, excluding the two respondents who did not provide a specific percentage of patients in their ideal world. We also asked providers to rate their satisfaction with the psychologist's communication, specifically regarding how the patient's emotional and behavioral health may affect their medical condition and/or treatment. Of the responses, 18.4% said that they do receive satisfactory communication, 65.3% said they sometimes receive satisfactory communication, and 12.2% said they did not receive satisfactory communication. Responses were not different based on center size $X^2(4) = 3.546$, $p = .471$ or reported percentage of patients they would like to be followed by psychology in an ideal world, $X^2(6) = 4.125$, $p = .660$. When asked about present access to Psychology services compared to an ideal scenario, the majority of nephrologists indicated that in an ideal scenario, psychologists would be available to a larger portion of their patients (Fig 3).

However, nephrologists also indicated that there is room to improve effectiveness of the psychosocial services to meet the needs of nephrology patients. Just under half (47.9%) of respondents indicated that their psychosocial services were either minimally effective or less than adequate (Fig 4). Of note, only respondents from centers without a dialysis unit indicated psychosocial services were minimally effective.

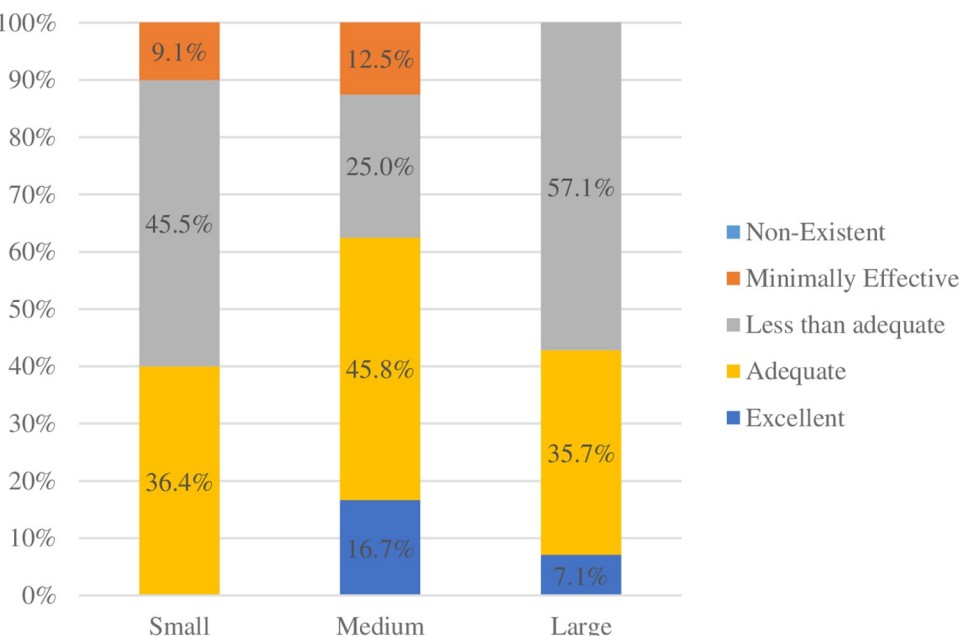

**Fig 4. Effectiveness of psychosocial services by program size.**

## Discussion

The current study sought to describe the current landscape of pediatric nephrology center psychosocial services, inclusive of availability and satisfaction for such services, via survey distributed among participants in the Pediatric Nephrology Research Consortium. Since the PNRC represents the majority of centers in the US, this data is likely representative of the status of pediatric nephrology psychosocial services in the entire country. Collectively, pediatric nephrology centers vary widely in size (as assessed by number of physicians and providers in each program), and roughly align with workforce differences across the country which have been described previously [25]. That is, the middle 50 percent of respondents work in programs with physician group size ranging from three to seven nephrologists, and 25 percent work within a group size of four nephrologists or less. Again, similar to previous workforce reports, the majority of nephrology programs care for the entire range of pediatric kidney conditions [25]; however, metabolic bone disease is a condition not ubiquitously addressed by nephrologists across programs. The majority of nephrology programs reported performing kidney transplants, although centers with small physician group sizes were less likely to do so. As to be expected, the more nephrology providers at a center, the more chronic dialysis patients were cared for at the center. Not every program had a renal dialysis unit, with less than half of smaller centers having a RDU and three quarters of medium and large centers having a RDU.

The current data demonstrate that psychosocial supports are generally available within pediatric nephrology centers in the United States. However, the degree to which psychosocial services are available to, or embedded within, nephrology programs vary. Specifically, the larger the group/center size, the more likely centers were to have a social worker and psychologist specifically assigned to the nephrology division. Centers that responded and were ranked in the top 40 pediatric nephrology programs by U.S. News and World Report were more likely to have a pediatric psychologist specifically assigned to their division, and thus better equipped

to provide additional support that may positively impact patient outcomes, including kidney transplant survival.

The most commonly assigned psychosocial professionals in pediatric nephrology centers are social workers, with 83% of mid-sized and 100% of large centers having dedicated social workers. These professionals often serve multi-faceted roles, including helping to address social issues for patients and families, screening for behavioral issues, and providing triage and/or direct counseling services. Centers were less likely to have embedded pediatric psychologists than pediatric social workers, although all centers had access to pediatric psychologists.

While having access to psychology, the majority of programs reported that one quarter or less (0–25%) of their patients are followed by psychology in some capacity. Only two (large) centers had a neuropsychologist specifically assigned to the division, whereas, some small and medium centers did not even have access to these professionals within their institution. Neuropsychologists assess cognitive and behavioral functioning that may be impacted by medical, psychological, or familial factors. A neuropsychological evaluation can help providers and families to understand specific cognitive deficits and to identify the most appropriate interventions to support the child with kidney disease. For example, understanding of a patient's neurocognitive profile may help inform how to structure patient education or expectations for medication adherence to match with the patient's cognitive abilities or literacy level. No centers had a psychiatrist assigned to their division, but almost all centers could access psychiatry via consultation. Unfortunately, similar to the workforce crisis in pediatric nephrology, there is a dearth and misdistribution of child psychiatrists available in the U.S. [38], so even if access to psychiatry exists, timely evaluations and access to ongoing follow-up care may be limited. Despite CMS requirements for regular behavioral health screening (specifically, depression and quality of life) among end stage kidney disease patients, close to 50 percent (43.9% - 47.9%) of the responding centers were *not* regularly screening for behavioral health indicators within chronic kidney populations. The inclusion of psychosocial team members may help to support required screening, as well as to support identification and targeted intervention for the most vulnerable nephrology patients.

When examining psychosocial services available in RDUs specifically, adjunctive supports varied widely and did not appear dependent on center size. Child Life specialists were the most common service available in RDUs and thus may represent basic standard of care needs for pediatric patients receiving chronic dialysis. Massage therapists were the least common, with only two large centers reporting they were available in their RDU. Therapeutic recreation and early intervention specialists were also uncommon. Considering the average number of school hours missed due to attending hemodialysis sessions, it was notable that school teacher services were not universally available. However, school teachers were more commonly a part of RDU care when responding centers were highly ranked (top 40) by the US News and World Report, and again may represent a key service provision for highly disadvantaged chronic dialysis patients.

Although availability of psychosocial supports varied, these supports appeared to be highly valued within pediatric nephrology centers, with 67.8 to 99.9 percent of centers reporting that psychosocial supports are either important or very important, with greater importance associated with larger nephrology group/center size. Although most nephrology programs indicated that few (less than a quarter) of their patients are followed by psychology, the clear majority expressed a desire (in an "ideal world") that more of their patients were seen by psychology with most indicating they would like more than half of their patients followed by psychology in some capacity. Despite indicating the importance of psychosocial supports and highlighting an increased need for psychology supports specifically, the perceived effectiveness of psychosocial supports meeting the needs of nephrology patients varied and was often not favorable. Of

note, the larger centers, with greater psychosocial resources, were more likely to indicate room for growth in order for psychosocial services to meet the needs of their patients (i.e., despite having access to more resources, the majority of larger centers reported psychosocial service effectiveness was "less than adequate"). This identified higher level of unmet need may reflect that larger nephrology centers see a high volume of patients with complex and chronic medical needs, whose health is also disproportionately impacted by key social determinants of health. These include but are not limited to their neighborhood and physical environment, educational level and health literacy, community and social support systems, economic instability, food insecurity, and overall poorer quality of life. Nephrologists, as well as the healthcare community as a whole, are identifying that these patient population characteristics require correspondingly higher psychosocial support. It may also reflect that as nephrologists become more familiar with the role of psychosocial providers and recognize the benefits (e.g., due to having a division assigned social worker and/or psychologist), they actually desire even more access to these services for their patients. It is also important to note, that centers that described psychosocial services as only "minimally effective" were all centers without an embedded RDU. Considering that CMS regulations mandate psychosocial support for patients, this may explain that association.

Based on the survey results, social work is the most commonly available psychosocial discipline in pediatric nephrology centers, which may help to explain the high levels of burnout experienced by these team members [30], especially when other psychosocial support is not present. Nearly half of nephrology teams at small centers and all of the large pediatric nephrology centers have an embedded social worker. When a social worker is embedded in pediatric nephrology divisions, they may become the default provider for all patient psychosocial needs, creating an underutilization of other psychosocial disciplines by nephrology centers. Nephrologists and their patients are better served by having access to a full range of psychosocial providers, minimizing the potential for individual provider burnout and promoting better patient outcomes.

Pediatric psychologists, who are doctoral level professionals (and often faculty within hospital systems), are well positioned to support the pediatric nephrology team as well as pediatric kidney patients. They are able to assist in identifying the most appropriate psychosocially informed medical care, directing evidenced-based interventions with patients and families, and supporting existing nephrologists and the other psychosocial team members. Given their extensive training in a broad array of treatment modalities and the intersection of physical and mental health, pediatric psychologists have unique expertise. Pediatric psychologists work to develop and implement patient interventions around adjustment to medical conditions, behavior plans to support adherence to medical regimen (including modifications for neurodiverse children), as well as promoting positive health behaviors and resiliency to prevent future injury or complications from medical conditions or comorbid psychosocial concerns.

The integration of interdisciplinary teams has been introduced as a standard of care to help meet the myriad of medical and psychosocial needs of pediatric nephrology patients [4, 39–41]. While prior research has established that interdisciplinary care can vary widely across centers with variable team structure [42], results from our study may help define differences across centers by number of nephrology providers and patients served. The opportunity to collaborate within an interdisciplinary psychosocial provider team may help maximize best outcomes, such as improvement in medication adherence and preparation for patient transition to adulthood [43]. Leveraging the expertise of multiple psychosocial disciplines may help alleviate the burden of care that falls on a limited number of pediatric nephrology providers.

Strengths of the current study include providing an updated tally of the landscape of pediatric nephrology centers in the U.S. as well as a first look at the availability of psychosocial

resources at these centers. Prior publications have examined psychosocial resources within pediatric dialysis units [40, 42], but no publications to date have offered a summary of psychosocial supports, degree of availability of supports within these settings, as well as evaluation of satisfaction with and effectiveness of these supports by nephrologists.

The current study is limited by the reliance on the nephrologist for self-reporting of psychosocial support resources and further lacks information from the perspective of the psychosocial providers themselves. It does not evaluate patient-related outcomes, specifically centered around key social determinants of health that we know impact this patient population [44–47]. Psychosocial supports at PNRC sites in North America that did not respond to the survey ($N = 38$) were not available for this project so may have skewed the results. However, responding PNRC centers were found to be similar to non-responding PNRC centers in terms of group size. Moreover, responding centers included the majority of top ranked pediatric nephrology centers. Finally, information regarding how psychosocial team members interact, service billing/reimbursement functions, and the percentage of time psychosocial team members are assigned to nephrology were not explored in the current survey.

Future research should evaluate how psychosocial support services impact our understanding of social determinants of health and in turn may reduce their deleterious impacts on the pediatric nephrology population. Outcome data should also be evaluated for determination of the impact that increased psychosocial support (specifically those that are embedded in the practice) may have in reducing acute healthcare utilization. With the use of targeted interviews, research should focus on pediatric nephrology centers that have integrated psychosocial services into their practice and help elucidate how these services were established (i.e., structure of time allotment/financial funding), identify the key psychosocial providers included in interdisciplinary clinics, and undertake cost analysis supporting this integration [42].

Future research should also explore whether psychosocial providers benefit from having a specialization in pediatric nephrology care, which addresses the complexities of disease outcomes and management, as opposed to a general psychosocial background. Emerging special interest groups and certifications highlight a role for nephrology-specific training for those who care for psychosocial aspects of patients with pediatric chronic kidney diseases. Research is needed to further assess the benefits of having a dedicated and embedded psychosocial provider in contrast to someone who may be available for consultation on an ad hoc basis. It is possible that the presence of embedded psychosocial providers, with time and expertise dedicated to this high-risk population, may result in improved preventative interventions for anticipated psychosocial concerns associated with treatment of the child's chronic disease, as well as improve nephrologist familiarity and comfort with engaging psychosocial services.

In conclusion, there is great variability in provision of psychosocial services and assignments of psychosocial professionals in pediatric nephrology centers in the US. Larger centers, with more pediatric nephrologists, are more likely to have access to a wider variety of psychosocial service professionals, and centers with a pediatric dialysis unit are more likely to have embedded social workers and psychologists. While psychosocial providers were available, the majority of respondents reported that <25% of their patients were followed by psychology, and the majority of pediatric nephrologists expressed desire for more psychosocial support for their patients. Much work remains to better understand the provision of psychosocial services for these patients at different centers and how these services are funded and utilized. Such work can help in formulation of key best practices in organizing and providing quality psychosocial care for children with kidney disorders.

## Supporting information

**S1 Dataset.**
(SAV)

## Acknowledgments

We would like to thank all PNRC members for their contribution to this work.

## Author Contributions

**Conceptualization:** Anne E. Dawson, Camille S. Wilson, William E. Smoyer, Neha Pottanat, Amy C. Wilson, John D. Mahan, Julia E. LaMotte.

**Data curation:** Anne E. Dawson, Camille S. Wilson, Julia E. LaMotte.

**Formal analysis:** Anne E. Dawson, Camille S. Wilson, William E. Smoyer, Neha Pottanat, Amy C. Wilson, John D. Mahan, Julia E. LaMotte.

**Investigation:** Anne E. Dawson, Camille S. Wilson, John D. Mahan.

**Methodology:** Anne E. Dawson, Camille S. Wilson, William E. Smoyer, Julia E. LaMotte.

**Project administration:** Anne E. Dawson.

**Writing – original draft:** Anne E. Dawson, Camille S. Wilson, Neha Pottanat, John D. Mahan, Julia E. LaMotte.

**Writing – review & editing:** Anne E. Dawson, Camille S. Wilson, William E. Smoyer, Neha Pottanat, Amy C. Wilson, John D. Mahan, Julia E. LaMotte.

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
