## [Decision Letter · Decision Letter 0]

14 Feb 2023

PONE-D-22-32765Psychosocial Supports within Pediatric Nephrology Practices: A Pediatric Nephrology Research Consortium SurveyPLOS ONE

Dear Dr. Mahan,

Thank you for submitting your manuscript to PLOS ONE. After careful consideration, we feel that it has merit but does not fully meet PLOS ONE’s publication criteria as it currently stands. Therefore, we invite you to submit a revised version of the manuscript that addresses the points raised during the review process.

We look forward to receiving your revised manuscript.

Kind regards,

Mohamed E Elrggal

Academic Editor

PLOS ONE

Journal Requirements:

   a. You may seek permission from the original copyright holder of Figure(s) [#] to publish the content specifically under the CC BY 4.0 license.  

Reviewers' comments:

Reviewer's Responses to Questions

**Comments to the Author**

1. Is the manuscript technically sound, and do the data support the conclusions?

Reviewer #1: Yes

Reviewer #2: Yes

2. Has the statistical analysis been performed appropriately and rigorously? 

Reviewer #1: Yes

Reviewer #2: Yes

3. Have the authors made all data underlying the findings in their manuscript fully available?

Reviewer #1: Yes

Reviewer #2: Yes

4. Is the manuscript presented in an intelligible fashion and written in standard English?

Reviewer #1: Yes

Reviewer #2: Yes

5. Review Comments to the Author

Reviewer #1: Thank you to Dawson et al for submitting this manuscript describing outcomes from a survey of paediatric nephrology centres across the USA in regards to psychosocial support services. It is well conducted and written. Thank you to the authors for the substantial work that has been undertaken. I only have minor queries.

1) It may be worth making mention of the diversity of stages and treatments of kidney disease amongst children as well as the etiologies, especially given that the burden of psychosocial support needs is likely different across the spectrum from early CKD to dialysis to transplant.

2) Whilst all of the questions I had in regards to the PNRC are subsequently answered later in the methods, it would be better to include some of these early in the manuscript when the PNRC is first mentioned.

3) More exploration of transition (ie, paediatric to adult services) in the context of psychosocial supports/services would be valued and increase the likely readership.

4) The discussion does feel quite long at times, but the content is highly valued. I would suggest a small table perhaps of the key take point points to help readers to engage with that content better.

Reviewer #2: Thank you for this manuscribt which can make changing in ouur practice review the writing .there is spelling and grammar need revision. we need abbreviation of PNRC not clear more detailed.regarding discussion add reference at end of each paragraph

6. PLOS authors have the option to publish the peer review history of their article (what does this mean?). If published, this will include your full peer review and any attached files.

Reviewer #1: No

Reviewer #2: No

---

## [Author Response · Author response to Decision Letter 0]

29 Mar 2023

3/31/2023

RE: Manuscript ID PONE-D-22-323765

Dear Dr. Elrggal, 

Thank you for your action letter received on 14 February 2023, which included the critiques from two reviewers for the manuscript, Psychosocial Supports within Pediatric Nephrology Practices: A Pediatric Nephrology Research Consortium Survey. My co-authors and I appreciate the reviews and the opportunity to revise this manuscript for resubmission. 

Below, please find our point-by-point responses, with the quote of each reviewer followed by our response. We have also attempted to highlight the substantive changes made within the body of the resubmitted document. We appreciate the encouraging comments from the reviewers that our manuscript was well conducted, written, and will help contribute to change in pediatric nephrology practices. We are thankful for the recommendations we received, and we hope you agree that this manuscript is now a stronger report. 

Journal Requirements

Reviewer comment Revise and resubmit response 

We referenced the style templates and adjusted our headings, figure/tables, and citations according to the template guide. 

2. Please provide additional details regarding participant consent. In the ethics statement in the Methods and online submission information, please ensure that you have specified what type you obtained (for instance, written or verbal, and if verbal, how it was documented and witnessed). If your study included minors, state whether you obtained consent from parents or guardians. If the need for consent was waived by the ethics committee, please include this information. Thank you for the recommendation to outline our consent process in greater detail. We have added text to the document that addresses the areas requested in our Methods, specifically that our research was determined by our institutions as not human subjects and consent was waived. 

 Thank you for the request to make more specific the ethics statement in our manuscript’s Methods section. As stated above, we have included more specific language that outlines the waiver that was granted by both institutions sponsoring the project. 

 a. You may seek permission from the original copyright holder of Figure(s) [#] to publish the content specifically under the CC BY 4.0 license. 

 Map images are in public domain searched through Google images. We switched to using a map that explicitly states the rights of use- for personal and commercial purposes, and do include reference to location map was found, per recommendation from the website: https://simplemaps.com/resources/svg-license

 We are providing a copy of our dataset in SPSS format (i.e., .sav), though we have deleted the variables identifying PNRC site- as we indicated to participants that this information would not be shared. 

6. Please review your reference list to ensure that it is complete and correct. If you have cited papers that have been retracted, please include the rationale for doing so in the manuscript text, or remove these references and replace them with relevant current references. Any changes to the reference list should be mentioned in the rebuttal letter that accompanies your revised manuscript. If you need to cite a retracted article, indicate the article’s retracted status in the References list and also include a citation and full reference for the retraction notice. Our reference list was reviewed and updated per additional citations included per recommendations of the reviewers. 

Reviewer 1

Thank you to Dawson et al for submitting this manuscript describing outcomes from a survey of paediatric nephrology centres across the USA in regards to psychosocial support services. It is well conducted and written. Thank you to the authors for the substantial work that has been undertaken. I only have minor queries.

1) It may be worth making mention of the diversity of stages and treatments of kidney disease amongst children as well as the etiologies, especially given that the burden of psychosocial support needs is likely different across the spectrum from early CKD to dialysis to transplant. We appreciate the support for this manuscript. Related to comment 1, we briefly highlight the heterogeneity in the care required by pediatric nephrologists in the first paragraph of this report, and we have added supporting citations. 

2) Whilst all of the questions I had in regards to the PNRC are subsequently answered later in the methods, it would be better to include some of these early in the manuscript when the PNRC is first mentioned. We added brief language to briefly introduce the purpose of the PNRC earlier in our introduction. 

3) More exploration of transition (ie, paediatric to adult services) in the context of psychosocial supports/services would be valued and increase the likely readership. We wholeheartedly agree with the reviewer regarding the importance of the potential role of psychosocial providers to be involved in transition to adulthood needs for kidney patients and their families, and we have made reference to that in the discussion. Per your recommendation, we have also highlighted our additional sentence in the introduction calling attention to this important consideration. 

4) The discussion does feel quite long at times, but the content is highly valued. I would suggest a small table perhaps of the key take point points to help readers to engage with that content better. Thank you for the suggestion of including a table of key take away points. We have included a summary of key findings/conclusions to help the reader see at a glance the main findings at the end of our discussion section. 

Reviewer 2

Reviewer #2: Thank you for this manuscribt which can make changing in ouur practice review the writing .there is spelling and grammar need revision. we need abbreviation of PNRC not clear more detailed.regarding discussion add reference at end of each paragraph

 The Pediatric Nephrology Research Consortium is initially spelled out and briefly explained on pg. 4. Due to request to clarify the abbreviation, we again spell it out at the start of the Method section and Discussion section, when PNRC is re-introduced. We have also thoroughly reviewed our manuscript and corrected typos/grammars throughout where we could identify potential concerns.

---

## [Editor Report · Decision Letter 1]

16 Apr 2023

Psychosocial Supports within Pediatric Nephrology Practices: A Pediatric Nephrology Research Consortium Survey

PONE-D-22-32765R1

Dear Dr. Mahan,

We’re pleased to inform you that your manuscript has been judged scientifically suitable for publication and will be formally accepted for publication once it meets all outstanding technical requirements.

Kind regards,

Mohamed E Elrggal

Academic Editor

PLOS ONE

---

## [Editor Report · Acceptance letter]

28 Apr 2023

PONE-D-22-32765R1 

Psychosocial Supports within Pediatric Nephrology Practices:
A Pediatric Nephrology Research Consortium Survey 

Dear Dr. Mahan:

I'm pleased to inform you that your manuscript has been deemed suitable for publication in PLOS ONE. Congratulations! Your manuscript is now with our production department. 

Kind regards, 

on behalf of

Dr. Mohamed E Elrggal 

Academic Editor

PLOS ONE